# Methodology for the Determination of a Process Safety Culture Index and Safety Culture Maturity Level in Industries

**DOI:** 10.3390/ijerph19052668

**Published:** 2022-02-25

**Authors:** Dorota Siuta, Bożena Kukfisz, Aneta Kuczyńska, Piotr Tomasz Mitkowski

**Affiliations:** 1Faculty of Process and Environmental Engineering, Lodz University of Technology, 90-924 Lodz, Poland; 2Faculty of Security Engineering and Civil Protection, The Main School of Fire Service, 01-629 Warsaw, Poland; bkukfisz@sgsp.edu.pl; 3Institute Safety Engineering, The Main School of Fire Service, 01-629 Warsaw, Poland; akuczynska@sgsp.edu.pl; 4Division of Chemical Engineering and Equipment, Faculty of Chemical Technology, Institute of Chemical Technology and Engineering, Poznan University of Technology, 60-965 Poznan, Poland

**Keywords:** process safety culture index, process safety culture maturity model, energy industry, sustainability

## Abstract

A mature of safety culture is crucial to preventing and mitigating accidents, incidents, and unsafe behaviors in the process industry, so as to make it more sustainable and economically responsible. Measurement, investigation, and assessment of the safety culture using interviews, questionnaires, behavior observation, reviewing documentation, and its impact on the safety performances of an organization is complicated, challenging, and requires a commitment to all employees. The aim of this study was to propose a novel, unique semi-quantitative methodology for the determination of a total process safety culture index and parametric model of process safety culture maturity in an organization based on the Bradley model. The methodology includes a questionnaire concerning different process safety culture factors, calculation procedures, and a graphical tool. In addition, three quantitative survey indicators were proposed: indicators of direct communication, average communication time, and the applicability rate of the proposed changes by employees. A fully-developed total process safety culture index allows for identifying, hierarchizing, and benchmarking different factors of the safety culture among companies and sectors. Moreover, it will enable identifying the area of actions required to improve safety practices and elements applied to the organization analyzed. The proposed methodology was verified in a case study of one energy company with three locations in Poland and can be easily applied to different industrial fields, including logistics and warehousing, the food industry, the paper industry, security services, fire services, and environmental and other agencies.

## 1. Introduction

The level of safety culture in industrial organizations plays a significant role in reducing or eliminating accidents, incidents, or near misses including human, economic, and material losses. Investigations of causes and consequences of historical events have revealed various factors influencing safety culture. Accidents in Ludwigshafen (1948) [1], Meda (Seveso, 1976) [2], and especially in Pennsylvania (1979) [3,4], Bhopal (1984) [5,6,7], and Toulouse (2001) [8,9,10] revealed the problem of lack of process knowledge and competence of operators, firefighters, and supervisors, lack of adequate operation procedures and documentation, poor monitoring of the operating conditions, and wrong plant location and layout in relation to the surroundings. The accident in Toulouse (2001) [8,9,10] highlighted the aspect related to lack of safety knowledge concerning the identification of hazard and accident scenarios as well as lack of emergency preparedness. Identical root causes were evident in the activities of Transocean and BP in the context of the accident at the Deepwater Horizon oil rig (2010) [11,12,13,14,15]. Emergency plans should include a number of worst-case scenarios, as their severity of consequence and dynamics are important, even if their probability is considered extremely low. After all, mitigation of risk at source must be aimed at reducing the potential hazard, the likelihood of accidents, limiting the consequences through appropriate organizational and technical systems [8,16,17,18], and by designing inherently safer processes [19,20,21]. Even in the power generation industry, which generally does not struggle with many different hazardous chemicals, accidents such as those in Chernobyl (1986) [22], Sayano-Shushenskaya (2009) [23], or Fukushima Daiichi (2011) [24] might be related not only to the technical but also managerial and safety culture issues [25,26,27]. In particular, these accidents present deficiencies in communication throughout the organization between different levels of managers, employees, authorities, and services responsible for public safety and external communities. The lack of a learning culture regarding incidents and accidents in similar installations, which could result in better identification of hazards, is obvious.

A relatively new aspect highlighted in some of the more current investigational reports is the presence of non-employees, most often contractors, during accidents at the plant, for example in Toulouse (2001) [8,9,10], Texas (2005) [28,29], and Macando (2010) [11,12,13,14,15]. At the outbreak in Toulouse in 2001, 266 plant employees and 100 representatives of subcontractors were present [8,9,10]. In Texas in 2005, the majority of the victims were employees of external companies found in offices located in close proximity to the installation commissioned [28,29]. On the other hand, the basic operation of the Deepwater Horizon drilling platform included Transocean employees and was only partly managed by BP employees [11,12,13,14,15]. These accidents represent deficiencies in effective process safety leadership, communication between actors involved in work performed, and learning culture.

As a result of the international audit conducted by NSOAF (North Sea Offshore Authorities Forum) in 2012–2013 in the North Sea oil production area, significant discrepancies between the operator’s and drilling contractor’s safety systems were identified, which means that this is an international problem, not just an individual problem which occurred between Transocean and BP [13]. In addition, BP has experienced several significant accidents related to process safety in various business segments over a decade, including Grangemouth (2000), BP Texas City (2005), BP Prudhoe Bay (2006), and Macondo (2010), which leads to questions on the quality and effectiveness of corporate safety management. At the time of the Macondo outflow, no independent BP board member had professional experience in oil and/or gas production drilling offshore. An analysis of BP board communications before and after the Texas City disaster in 2005, as well as BP and Transocean communications before and after the Macondo disaster, illustrated the evolution of approaches to process safety and communication for major accident prevention from the BP board’s perspective and presents a rather more static and traditional approach than that of Transocean [13]. After the accident at Macondo, the Maritime Safety Authority and the Bureau of Safety and Environmental Enforcement (BSEE) issued guidelines on best practices for the maritime industry in creating an effective safety culture [30]. This is the first document in the history of that organization regarding safety culture. This statement defines a culture and safety policy but does not deal with the role of the board of directors in creating a strong safety culture, leaving this issue to the leaders. It should be noted here that the American Institute of Chemical Engineers published its guidelines on aspects of safety culture much earlier [31].

In summary, these accidents highlighted deficiencies in communication within the same organization, between companies (such as owners and installation operators) and external communities, lack of employee learning culture, low commitment of people working for these companies, neglect of good engineering practices, insufficient safety management systems, non-incorporation of process safety into the management decision process, and lack of leadership and process safety knowledge and competence. All these aspects can be placed in one bucket called the safety culture. Taking a close look at the root causes of accidents, incidents, and near misses in the process industry can reveal that the low level of safety culture is the bottom line of almost all reported accidents.

Safety culture in relation to organizations and companies in energy industry has many definitions. As early as 1993, the British Health and Safety Executive (HSE) [32] coined the term “safety culture” to mean the result of recognized individual and group values, attitudes, perceptions, competences, and behavior patterns that determine the commitment, style, and proficiency in managing health, occupational hygiene, and safety in the organization. Organizations with a positive safety culture are characterized by communication based on mutual trust, a common perception of the importance of safety, and trust in the effectiveness of preventive measures. In contrast, Cox and Flin in 1998 [33] gave the definition “safety climate”, which is considered as a surface feature of a safety culture seen through the prism of attitudes and the perception of the workforce at a given moment. At this point, it is important to draw the reader’s attention to wider definitions related to organizational culture and climate. According to Schein [34], culture somehow implies rituals, climate, values, and behaviors that tie together into a coherent whole understanding as culture. In that perspective, Guldenmund [35] reviewed safety culture and climate literature and concluded that it is rather possible to determine safety climate while full assessment of organizational culture might be illusionary and dependent on a given point in time [36,37]. More details about the relationship between safety culture and safety climate can be found in the scientific literature [38,39,40].

In 2005, the HSE indicated five factors that influence safety culture [41]. These are leadership, two-way communication, employee involvement, learning culture, and attitude towards blame. Adoption of specific indicators, and thus their level of sophistication, may translate into the adoption of a specific maturity model of safety culture. In 2017, Markowski [42] defined safety culture as the total performance of employees, characterizing their behavior, ways of solving problems, and activities in the area of safety assurance that function and are recognized jointly in a given industrial organization or a given country. Later, in 2018, Goncalves and Waterson [37] observed that there is a vague definition of safety culture in the literature, and there is even an unclear distinction between the concept of “safety culture” and the concept of “safety climate”. It might be concluded that safety culture usually refers to underlying assumptions and values that guide behavior in an organization, rather than the direct perception of individuals in that organization.

For the purposes of this work, “safety culture” is defined as a set of values, conditions, procedures, and behaviors recognized both individually and collectively in the organization under consideration, regarding the organization of a management system to prevent and protect against errors, incidents, breakdowns, cyber-attacks, system integration, and accidents, and to promote safety-oriented behaviors between cooperating organizations in normal and emergency situations. This definition applies to employees of the organization and persons performing work commissioned by that organization. It is to be noted that an organization can be either monocultural (characteristic for rather small organizations) or multicultural, which might have a significant impact on the behavior of people responsible for creating that organization, especially regarding blame and perception of safety. Moreover, an organization can be business-oriented, non-profit, or a fire and emergency service, either local or national.

Safety culture can be actually treated as an essential part of the corporate culture, therefore playing an important role in strategic management and intelligence within the organization [43]. Strategic intelligence cannot exist without advanced operational and tactical intelligence. Without strategic intelligence, it is difficult to address issues related to the sustainability development in industry. Despite which organization’s intelligence is considered, either coming from the energy sector or not, it requires measurable data produced and obtained within an organization as well as its outside, such as economic, political, technological, environmental, and social and [44]. Therefore, qualitative assessment of the safety culture should be considered as a key future in the development of agile and resilient organizations, which are based on employee competence and promotion of employee involvement and distributed leadership. In that view, the safety culture is considering behavior and interactions of people creating and using the technology, and do not directly consider the aspects of technical safety itself.

The above short analysis of major accidents clearly shows and identifies parameters influencing safety culture. Therefore, this has motivated the authors to provide a comprehensive, although brief review, of safety culture models (discussed in Section 2), which have led to the development of a safety culture model based on the Bradley model (i.e., DuPont model [45]) resulting, in turn, in the development of a novel semi-quantitative methodology for determining a total safety culture index (shown in Section 3). The model with its developed methodology allows the comparison of different safety culture aspects within the whole or well-defined parts of the organization and the benchmarking of different organizations. Moreover, it allows the actions required to improve safety culture to be identified. This method can be used in the assessment of safety culture and relevance of various interactions between business-oriented organizations and safety services, such as fire services. This might have great importance during a rescue when access to buildings and process installations for the fire protection service is crucial.

## 2. Crucial Safety Culture Maturity Models

Application of the safety culture definition implies a need for a model definition. In the literature, safety culture is modeled in terms of safety culture maturity. Models of safety culture maturity define specific states or levels that assess the completeness of the objects analyzed, usually an organization or process, through various sets of multidimensional criteria [37,46,47,48]. Assessment, whether qualitative or quantitative, of a safety culture requires the identification of characteristics, and further adoption of indicators that will allow the assessment of the level of safety culture. Many quite simple and complex frameworks, models, and tools for qualitative and semiquantitative assessment of the safety culture have been proposed in the open literature [49,50,51,52,53,54,55] that can be used for assessment of safety culture in whole organizations as well as in independent driven groups within these organizations (such as task-oriented groups, research groups in academic institutions, and security services). In addition, they are related to personal safety and do not consider aspects of process safety.

Sutton [49] stated that there is a strong correlation between root cause analysis and the development of company culture. In paper [50], Sutton described safety culture as a function of people and management systems, which can be understood as the quality of the staff and quality of the management system. In one of the culture matrices, two features of culture were correlated, personalities and systems, and four states of cultures were distinguished: chaotic, tribal, bureaucratic, and operational excellence. Markowski [42] listed low, local, bureaucratic, and high cultures. Low culture is characteristic for organizations in which employee behavior and action are accidental and diverse, and often instinctive in nature. This is due to the low quality of the management system, which is not effective in setting behavior standards and enforcing them, nor in supporting employee development. The management system is erratic and unstable. Local culture refers to small organizations, which base their operation on highly qualified employees (professionals) who feel a strong relationship with the company and with each other. Due to the level of employee involvement, ease of communication, but low level of formalization, the management system in such organizations is relatively weak. In turn, bureaucratic culture is characteristic of larger and often older organizations where an efficient technical and organizational system is necessary for the efficient functioning of the company, but employees work in this system in an automated, bureaucratic manner. Despite all these aspects, the highest manifestation of distinctive culture is the involvement of qualified and motivated employees who are guided by the assumptions of a strong management system and in their conscious action ensure continuous improvement of the management system, striving for the so-called operational excellence.

Another closely related trend is based on behavior-based safety (BBS) programs, which are well propagated by DuPont through the Bradley curve [45]. The Bradley curve distinguishes four states of safety culture: reactive (safety driven by natural instincts), dependent (safety driven by management and based on supervisor control), independent (driven by personal knowledge, commitment, and standards), and interdependent (driven by care for others and team working). This way of modeling safety culture is well suited to the task of moving an organization from the most underdeveloped state to the interdependent phase. However, it is notable that for cultures that achieve sustained excellence, the BBS program alone is rarely enough, and some cases even suggest that prolonged use of the BBS program decreases effectiveness over time. In response, DuPont proposed the DNA strategy (the DuPont Integrated Approach for Safety) [45] which uses the felt leadership model to initiate and support significant and lasting cultural changes in all stages of development, regardless of whether the organization is just starting its journey to a sustainable safety culture or whether something new is sought. That method is based on control lists.

In 2001, Fleming [51] presented a list of 10 elements defining the maturity model of a safety culture. These are (1) management commitment and visibility, (2) communication, (3) productivity versus safety, (4) learning organization, (5) safety resources, (6) participation, (7) shared perceptions about safety, (8) trust, (9) industrial relations and job satisfaction, and (10) training. These elements are reflected in the five-level safety culture maturity model, starting from the emerging level and increasing through managing, involving, and cooperating levels up to the continually improving level. Filho et al. [37] went further in the development of the Fleming model [52] and used in their work the following components in the assessment of safety culture: (1) information on accidents and incidents (unusual events), (2) the organization’s ability to learn, (3) employee engagement, (4) communication, and (5) commitment of leaders.

In 2003, Hudson [53] proposed a safety culture maturity model consisting of five major stages: pathological, reactive, calculative, proactive, and generative. Organizations at the lowest level are mainly driven by business goals and a desire not to get caught by the regulator. At the reactive stage, organizations take safety seriously but there is only action after incidents. When safety is driven by management systems, with extensive collection of data imposed by management rather than looked for by the workforce, the organization is at the calculative stage. The proactive stage is characteristic for organizations in which the workforce is becoming involved, and initiative is moving away from a purely top-down approach. When active participation at all levels of organizations is observed, then the generative stage is tackled. For that step, it is characteristic to perceive safety as an inherent part of the business. That model has been used in health care [53], the aviation industry, and multinational organizations [54].

## 3. Proposed Model and Semi-Quantitative Methodology of a Process Safety Culture Determination

A new, unique, semi-quantitative methodology for determination of a process safety culture index was developed and is presented in Figure 1. It consists of a number of elements appropriate for different process safety culture beams or parameters, calculation procedures including equations for determining unique indicators of direct communication (*DI*), average communication time (*ACT*), and applicability rate of the proposed changes (*ArC*), and indexes (safety culture index (*SCI_i_*) and total safety culture index (*TSC*)), and a graphical tool. The proposed methodology allows to compare different safety culture aspects within the whole organization or its well-defined parts and to benchmark various similar or/and cooperating organizations. The methodology of the process safety culture index consists of nine major steps.

The first step is to define beams, which are process safety culture parameters that can be analyzed in the organization, such as personal knowledge and skills in the field of process technologies and safety, networking, communication, and information flow between leaders, workers, and local community under normal and emergency conditions, recognition and rewards, safety leadership of the corporate management, organizational pride, costs of proactive actions, care for others, personal care, inherent safety, safety training, personal injuries due to incidents and accidents, loss of working time, cost of incidents and accidents, loss of organization image, and process safety measures. These parameters can be grouped into two clusters: positive driven beams (PDB) and negative driven beams (NDB). In case of the positive driven beams, the following parameters are included: care for yourself, personal knowledge and skills in the field of process technologies and safety, networking, communication and information flow between leaders, workers, local community under normal and emergency conditions, recognition and rewards, leadership of the corporate management in process safety, organizational pride, costs of proactive actions, care for others, inherent safety, process safety measures, and safety training (inside the organization and with other organizations, e.g., fire brigades).

Opposite to PDB, the negative driven beams comprise personal injuries, loss of working time, number of casualties, loss of image, and cost of incidents and accidents. It should be noted that these PDB and NDB process safety culture parameters are simply examples and do not include all the possibilities. This step is highly dependent on the experience of the group and the subject of the analysis, as well as the specific client requirements. In the following step, the beams are divided into four groups: qualitative positive driven beams (2a1), quantitative positive driven beams (2a2), qualitative negative driven beams (2b1), and quantitative negative driven beams (2b2). In our perspective quantitative, beams lead to purely quantitative data such as the number of accidents, failures and to other quantitative indicators, such as those proposed in this study: indicators of direct communication, average communication time, and the applicability rate of the proposed changes by employees, which are explained in more detail in the following paragraphs.

In step 3, detailed questions regarding safety culture parameters are defined from the perspective of the identified beams. This step is highly dependent on the expertise of people constructing the questionnaire and on the investigated organization. The validity and reliability of the questionnaire should be tested. During the creation of the questionnaire, it should be assumed that, to a high extent, the safety culture means the quality of perception of safety within the organization. The perception of the same situations by people at different levels in the organization may differ. The questionnaire is directed to all employees within the analyzed organization, regardless of their position and functions. The questionnaire allows to unambiguously assign the answers to the position in the organization, and thus it is further possible to assess the quality of safety culture at a given organizational level.

In step 4, a survey is conducted and analyzed. In our case, we decided to use online questionnaires since this allows respondents to answer at any time and, if needed, to return after a break. With our experience, we foresee the need for some backsteps in the proposed methodology since there is no perfect set of model parameters or questionnaires.

Next, in step 5, the three new quantitative indicators of the survey are proposed and determined: the indicator of direct communication (*DI*), the average communication time (*ACT*), and the applicability rate of the proposed changes (*ArC*).

The indicator of direct communication in the organization structure describes whether there is a possibility of effective communication between employees and the management (also the owner, head, or top executive) of the organization and is defined by Equation (1). In practice, *DI* is defined as the inverse of the number of supervisors/persons (*ip*) who necessarily need to be officially notified in order to reach the head director (or decision-maker), if the case concerns process safety.
*DI* = *(ip)*^−1^,(1)

The indicator of average communication time (defined by Equation (2)) represents the average response time of the senior management or owner to incoming communication from the employee (or the person working for the organization) regarding process safety in the organization and is defined by response time (*tp*). In other words, this is the response time notification of the management to the employee expressed in days.
*ACT* = 2/(2 + *tp*),(2)

The applicability rate of the proposed changes (*ArC*, defined by Equation (3)) shows the ratio of the number of changes introduced and proposed by employees to the number of all changes reported by committees or other communications either in digital or physical documents.
*ArC* = *iC/iNC*,(3)

The *iC* stands for the number of changes introduced and proposed by employees (or persons working for the organization), while *iNC* represents the number of changes notified by all employees. It should be noted that changes proposed by persons responsible for safety should be reported separately because that is a part of their basic work duty.

For the purposes of quantitative analysis of the questionnaire, it was assumed that the Bradley curve can be described by an exponential function of base 2. Therefore, the safety culture level (*SL*) is expressed as a function of the parameter representing the advancement of safety culture level (*ASL*) and is represented by Equation (4).
*SL* = 2*^ASL^*,(4)

In step 6, the quantification of the questionnaire is performed under the assumption that each subsequent answer to a given question corresponds to the specific level of the safety culture. Therefore, considering that *y_i_* corresponds to each subsequent answer *I* in a given question *y*, this yields the sum of the value of answers divided by the maximum values of answers to individual questions and gives the individual safety culture index of the given respondent *r* (*SCI_r_*), which is represented by Equation (5).
(5)$$SCI_{r}=\frac{\sum_{z=1}^{n}y_{z,i}}{\sum_{z=1}^{n}\left(\max y_{z,i}\right)},\;z\in n,$$

It should be noted that *y_i_* in Equation (5) stands for the value of answer given in the questionnaire by respondent *r* in the *i*-th question, and *n* represents the number of all questions in the questionnaire. The particular value of answer *y_i_* is defined as the value of the following answer *i*, which can take values depending on the total number of possible answers *k* to a given question *i* according to Equation (6). Considering Equation (6), Table 1 was created to show the values of individual answers depending on the number of possible answers.
(6)$$y_{z,i}^{}=\frac{2^{i}}{\sum_{j=1}^{k}2^{j}},\;j\in k,$$

Step 7 of the proposed methodology consists of the calculation of the total process safety culture index (*TSC*) at a given employment level (*TSC_@E_*) according to Equation (7). The index is calculated under the assumption that *r* is the number of all responders at the investigated level. *E* denotes the given employee level in the organization or for the whole organization, or it is associated with an external entity (tires). It should be noted that if *r* is used to represent all employees in the organization, it will provide the value of the total level of the process safety culture index of the investigated organization.
(7)$$TSC_{@E}=\frac{\sum_{i=1}^{r}SCI_{i}}{r},$$Within the presented methodology, we proposed the assessment of the beam process safety culture (*SCI_b_*) by means of averaging sum of *SCI*_i_ of each respondent *r* with respect to only *n* questions, which are related to the specific beam (see Equation (8)).
(8)$$SCI_{b}=\frac{\sum_{j=1}^{n}\left(\frac{\sum_{i=1}^{r}SCI_{i}}{r}\right)}{n},$$

It should be noted that *SCI_i_*, *SCI_b_*, and *TSC_@E_* can vary between 0 and 1, where values approaching 0 indicate the lowest possible level of safety culture, and 1 is the ideal safety culture. 

In step 8, the safety culture maturity level in the organization is determined based on the ranges of values of *SCI_i_, SCI_b_*, and *TSC_@E_*, and assigned according to the ranges shown in Table 2.

The proposed parametric model of process safety culture maturity based on the Bradley model consists of four levels of safety culture, which are described in Table 2 and presented in Figure 2. The lowest level of safety culture is named reactive and forms the center of the model in Figure 2. The next levels are dependent, independent, and interdependent. There is a set of model safety culture parameters that are formulated in Figure 2 as positive and negative driven beams (PDB and NDB, respectively). Maturity of the safety culture concerns a desired increase of the PDB values, while in case of the NGB, a decrease to the lowest possible values is preferential.

The colors represent the levels of safety culture maturity and are intentionally fuzzy; there is no clear border because from the authors’ perspective it is difficult to define a border between each level of maturity. In real cases, some of the beams can be at a higher level of maturity while others at a lower one. Therefore, the model postulates that an organization can be at different levels of safety culture maturity depending on which parameter or set of parameters are assessed. Considering the parametric model of safety culture maturity, it is possible to distinguish extreme shapes of safety culture maturity for interdependent and reactive levels, which are presented schematically in Figure 3. For the interdependent level, the safety culture measure forms a canopy-like shape (Figure 3a), while for the reactive level becomes a bib-like shape (Figure 3b). The organizations under transformation typically form an expanding canopy (bubble-like) moving towards the full canopy-like shape.

The last step (9) concerns the analysis of the parametric model of safety culture maturity in a given organization and the identification of process safety areas that should be improved. The strong beams will be represented by the highest values of *SCI_b_*, while the beams with lower *SCI_b_* will require improvement within the analyzed organization.

## 4. Case Study

The proposed methodology for determining the process safety culture index was tested and validated in three different sites of one organization related to the energy industry, namely “L”, “K”, and “G”. The personnel structure of the sample in the analyzed organization comprised top executives (2.30%), managers (8.90%), administration employees (23.10%), operational staff (56.80%), and independent professionals (8.90%). The age of the personnel ranged from 19-73 years (M = 31.69, SD = 9.88). The majority of respondents had secondary education (39.77%), 35.27% of respondents had higher education, 21.10% had professional education, and 3.86% had primary education. For this study, 14 beams were identified based on the authors’ experience and agreement with the organizations’ representatives, namely leadership for safety, recognition and awards, communication and information flow between leaders, workers and local community under normal and emergency conditions, networking, personal knowledge and skills in field of process technologies and safety, care for yourself, loss of organization image, fatal accident rate, loss of working time, personal injuries, safety training, care for others, and cost of proactive actions and organizational pride. The process safety culture questionnaire consisted of 142 closed questions; however, due to the company’s sensitivity data only the selected questions are presented in Appendix A. The questionnaires were sent to the employees and a total of 528 valid completed questionnaires were gathered (186—site L, 175—G, 167—K). The proposed questionnaire presented good reliability and internal consistency for all the scales and dimensions with Cronbach’s alphas [56] ranging from 0.71 to 0.82 (see Table 3). The values of Cohen’s kappa coefficient reached a good repeatability of the responses [57] (Cohen’s kappa coefficient: 0.62—0.80).

## 5. Results and Discussion

Results of the process safety culture questionnaire for each site and beam were analyzed and shown in Figure 4 for the first time, at least to our knowledge. The beam of care for yourself obtained the highest *SCI_b_* values in each site. The site G reached only in that beam the highest value while in the rest of safety culture beams have mainly lowest values in comparison to other sites within the organization. It is especially surprising in view of the lowest value of the safety training beam, which is clearly at the reactive level of safety culture maturity. This is in good agreement with low *SCI_b_* values of personal knowledge and skills in the field of process safety technology. Only three beams are fully at the independent level, namely leadership for safety, networking, and care for yourself, which in our opinion makes a very good foundation for levering all sites and respective beams to the higher safety maturity level. Improvement in terms of safety training and personal knowledge can be achieved with the creation of an open-minded environment for process safety meetings, an increase of lessons learned from analysis of near misses, incidents, and accidents. Safety training should not focus only on personal safety and basic operational issues but also on process simulations emulating near misses or emergency cases with the use of digital twins (virtual training for emergency situations). Process simulators improve skills of the control room staff, especially responding to emergency situations in near real-life conditions; the same should be conducted for operators working at installations, as well as firemen. Safety and security issues of industry 4.0 technologies are also challenging. Therefore, gaining of new knowledge and skills is desired by employees in the field of process safety related to full automation and mechanization of processes, machine, and equipment diagnostics to ensure reliability and integrity of process operations.

The novel quantitative indicators of the safety culture survey, such as direct communication, average communication time, and applicability rate of the proposed changes, were determined and are summarized in Table 4. The direct communication indicator (*DI*) is quite similar for sites L and K (ranging from 0.41 to 0.44) with a quite distinctive value of 0.33 for site G. Moreover, the *DI* values are similar in groups of managers, operational staff, and independent professionals, which either suggests that safety communication is directed to more than two decision-makers or it shows that the organization is taking safety decisions in groups or teams. This can be supported when noticing the results for the operational staff of site G, where *DI* is the smallest (*DI* = 0.28), suggesting more persons need to be informed about safety issues but at the same time pointing to a high applicability rate of proposed changes (*ArC* = 0.8). On the other hand, *DI* values for administrative staff are higher than for operational staff, at the same level or close to the level of independent professionals. That suggests that safety issues are discussed in smaller groups or limited to the closest superiors. In site L, the involvement of employees in reporting and considering proposed changes in the field of process safety should certainly be strengthened, since it is lower than for site G but almost at the same level as in site K. An interesting observation is made for the group of top executives for which *DI* reached values of one. That might be related to the fact that safety issues are discussed with other top executives or with decision-makers. At the same time, the application rate (*ArC*) is below 1, which can be due to not all proposed changes by top executives being implemented. It suggests that either insufficient resources were assigned or that the decisions were made by groups that can reject ideas coming from top executives.

The indicator representing the average communication time (*ACT*) ranges for all sites between 0.24 to 0.32 but with a rather high variation between groups of different positions. It seems that *ACT* represents a relatively low value of 0.27 for the whole organization, which might suggest that the organization needs time to proceed with actions aiming at safety improvements. *ACT* can be back calculated to a number of days, which for a value of 0.27 results in an average response of 5.4 days. The direct communication indicator reached a value of 0.4, which is a good result, suggesting that decisions about safety do not require a long hierarchical way to be taken. However, there is a pattern that in each group independent professionals have rather high values in comparison to other groups, suggesting that professionals have a faster track to decisions than others. At the same time, top executives have lower values than professionals, which can be explained by the longer time required to make a decision either by boards of top executives or the requirement of collecting more data due to more complicated cases.

The summary of results presented in Table 4 brings one surprise. Groups of top executives have one of the lowest values in the total process safety culture index varying between 0.44 and 0.63. That can be explained by the fact that not all top executives are technically oriented, which makes them less process safety oriented. It also might suggest that some of the top executives are specialized in different business areas.

In general, the organization is positively oriented towards process safety which is manifested by the *TSC_@O_* equal to 0.53 (see Table 4). It does not only fulfil the industry standards but also involves their employees in the improvement of process safety in their organization. Site L obtained the best among all groups and averaged a total process safety culture index (*TSC_@L_*) of 0.56, which corresponds to the independent level according to the used scale (the same level obtained site K (*TSC_@L_* = 0.54)). By contrast, site G obtained a slightly lower *TSC_@G_* value of 0.49, which corresponds to the dependent level. However, the average value of the total process safety culture index for all sites is equal to 0.53 ± 0.035 which makes the organization convergent regarding the independent safety culture level.

The overview of the results obtained from the safety culture study covering the whole organization is presented graphically in Figure 5. Leadership for safety, networking, and communication and information flow between leaders, workers, and local community under normal and emergency conditions are those positive driven beams of the parametric model of safety culture maturity, which drive an organization into an independent level of safety culture. That is supported with extremely low values representing the fatal accident rate, loss of working time, and personal injuries. Two parameters that bring down the safety culture are safety training and process knowledge and skills in the field of process safety technology. Therefore, an improvement within these two hotspots is a must in order to fully achieve the independent safety culture maturity level, while improvement of the organization’s safety culture requires development within all areas.

The improvements should consider careful preparation of safety training oriented on the utilization of lessons learned from accidents, incidents, and near misses that shape the safety culture. Historical data provide extensive information about the root causes, conditions, and consequences of failures to protect installations against similar events. The transfer of experience and open information between organization sites increase the level of knowledge and skills that could not be achieved without such lessons. This allows for a more accurate hazard analysis and risk assessment, and on this basis, the development of an effective safety management system. The ability of plant employees to learn from the mistakes of others is an expression of interdependent safety culture. Additionally, employees should improve their knowledge and skills in the novel field of process safety of industry 4.0 technologies related to full automation of processes, use of big data, cloud computing and cyber safety. A good sign of future progress is related to a very good level of leadership (mainly in the managerial group) and care for yourself, although a disturbing hotspot is to be found on the safety culture maturity level of top executives (see Table 4). Nevertheless, top executives (especially in site G) generally express their wishes for safety improvement and show their leadership, which is also reflected by managers who are driven towards process safety improvement. That gives reasonable hope for a better, safer future for the entire organization.

## 6. Conclusions

In the study, a novel, unique semi-quantitative methodology for determination of a total process safety culture index and parametric model of safety culture maturity in an organization based on the Bradley model was developed and presented. The methodology uses a questionnaire aiming to investigate various safety culture factors, calculation procedures, and a graphical tool. In this work, three quantitative indicators assessing direct communication, communication time, and applicability rate of the proposed changes by employees were proposed and investigated on the test case study, along with a quantification tool for qualitative answers. The results of the quantitative answers are easily translated with the use of the proposed Equation (6) under the assumption of the safety culture maturity level represented by Equation (4), and ultimately lead to the calculation of safety culture indexes and total process safety culture index.

The presented case study shows applicability of the total process safety culture index in the energy industry. The methodology can be used to show the safety culture hotspots, which need improvement within the whole organization and specific position levels. It is important to point out that the obtained results of safety culture are for a given point of time and experience of top executives and managers in process safety. To see in which direction, the organization and their specific sites are going to, the safety culture assessment should be carried out over the distinctive periods of time with the same methodology and criteria. Such assessments with conjunction of analysis of specific beams could provide important results about behavior of the safety culture in organizations.

The model can be easily extended and adapted to specific requirements and an organization’s environmental conditions due to the unique experience or expertise of its safety culture investigators. It can be applied to various fields, including logistics and warehousing, the food industry, the paper industry, fire services, and environmental agencies. The use of the proposed model with its quantification of quantitative and qualitative data requires more real-life studies focused on creating a set of benchmark cases.

## Figures and Tables

**Figure 1 ijerph-19-02668-f001:**
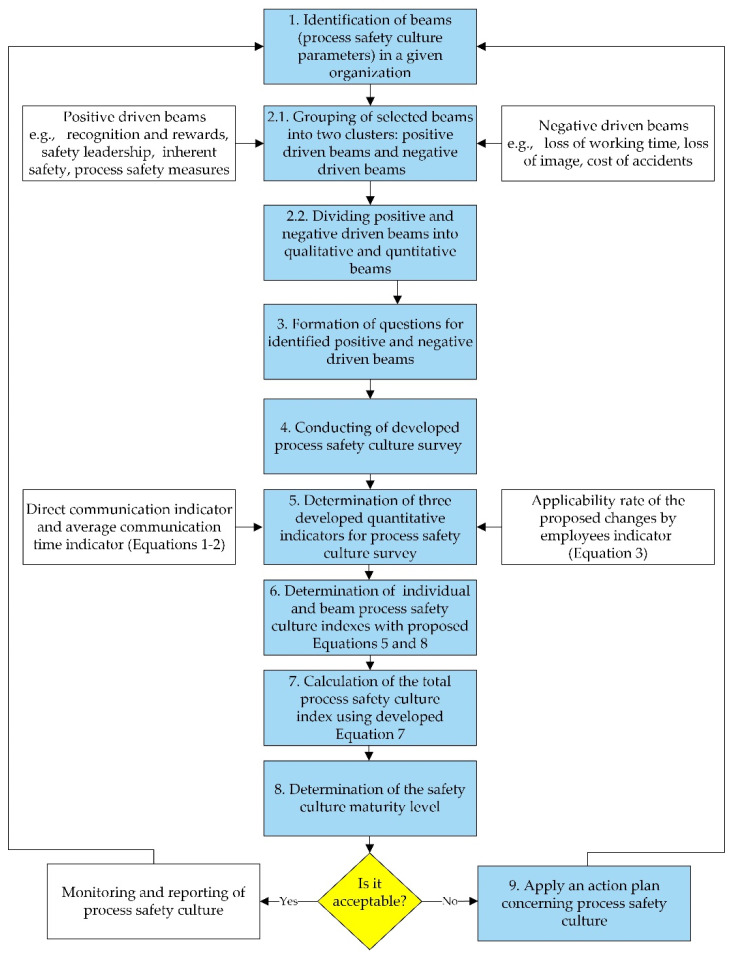
Methodology for determination of process safety culture index.

**Figure 2 ijerph-19-02668-f002:**
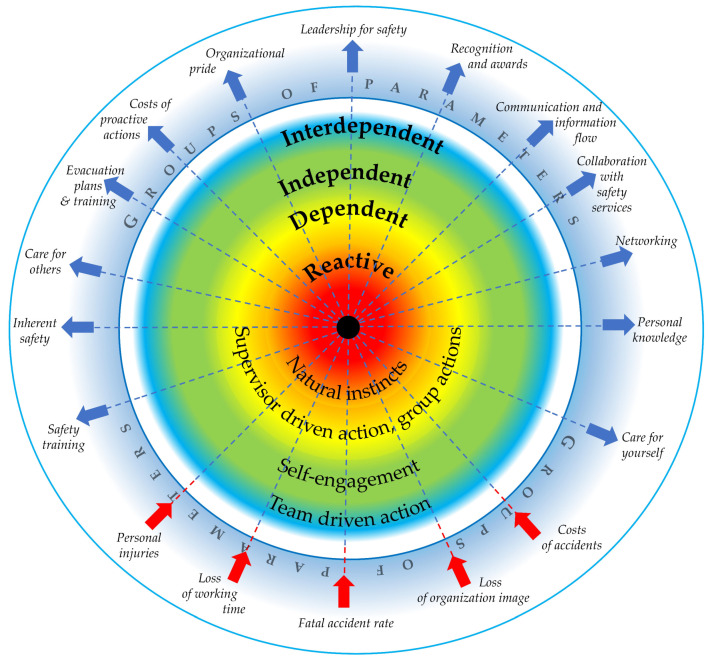
Parametric model of safety culture maturity.

**Figure 3 ijerph-19-02668-f003:**
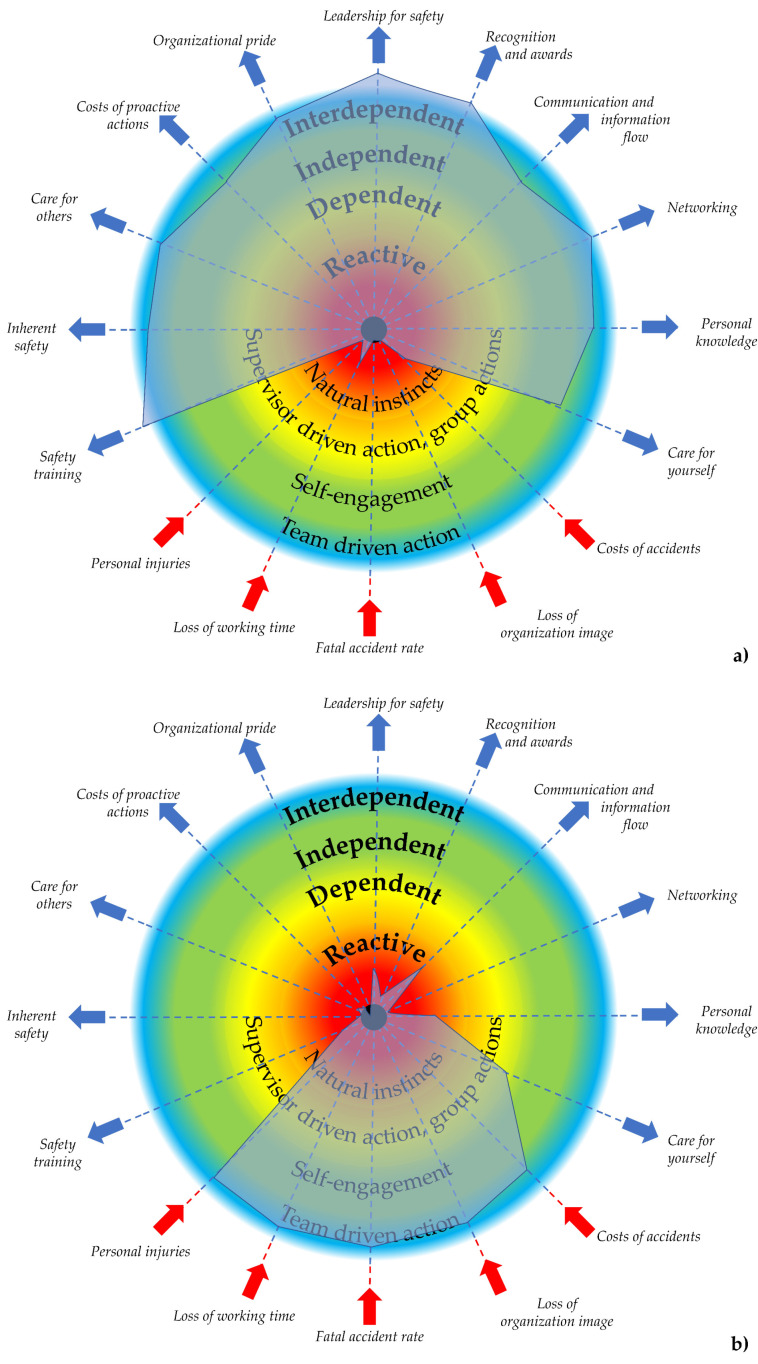
Characteristic shapes of safety culture maturity: (**a**) canopy-like shape—independent level of safety culture maturity, (**b**) bib-like shape—reactive level of safety culture maturity.

**Figure 4 ijerph-19-02668-f004:**
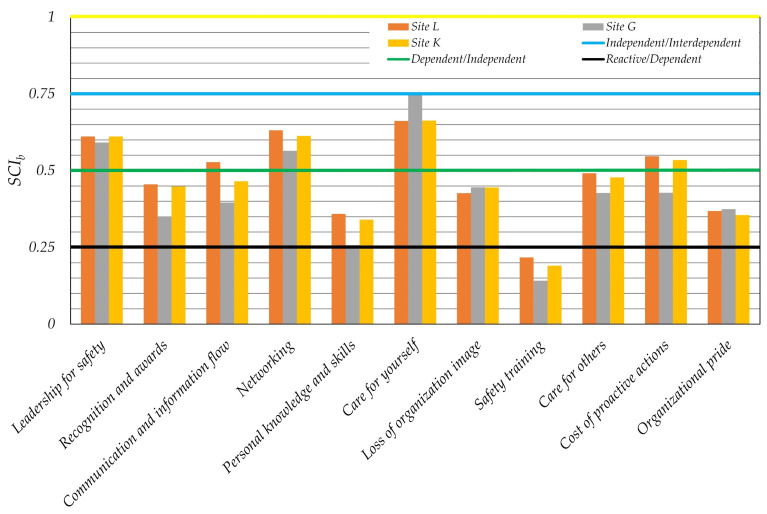
Process safety culture index for each qualitative beam and three sites with borders of safety culture maturity levels.

**Figure 5 ijerph-19-02668-f005:**
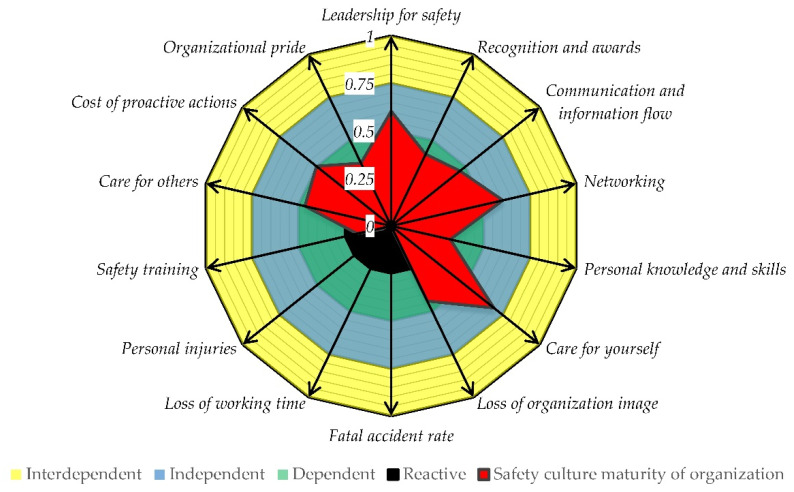
Shape of safety culture maturity in the whole organization of the energy industry.

**Table 1 ijerph-19-02668-t001:** A proposal for quantitative assessment of the questionnaire regarding process safety culture taking into account Equations (5) and (6).

Number of Answers *k* in Question *z*	Following Answers *i* in Question *z*	Value of Answer *i* in Question *z* (*y_z,i_*)	Sum of All Values of Answers in Question *n* $\sum_{i=1}^{k}y_{z,i}$
2	1	0.33	1
	2	0.67	
3	1	0.14	1
	2	0.29	
	3	0.57	
4	1	0.07	1
	2	0.13	
	3	0.27	
	4	0.53	
5	1	0.03	1
	2	0.06	
	3	0.13	
	4	0.26	
	5	0.52	
6	1	0.02	1
	2	0.03	
	3	0.06	
	4	0.13	
	5	0.25	
	6	0.51	

**Table 2 ijerph-19-02668-t002:** The level of safety culture and the adopted values of *SCI_i_*, *SCI_b_*, and *TSC_@E_* indexes.

Maturity Level of Safety Culture	Description	Range of Values for *SCI_i_*, *SCI_b_* and *TSC_@E_*
Interdependent	Team driven safety actions. Creation of best practice. Safety drives all goals in the organization. The organization motivates others to improve safety.	〈0.75;1〉
Independent	Safety aspects well-known to all members of the organization. Adoption and attention to best practices. The organization includes safety in cooperation with third parties.	〈0.50;0.75〉
Dependent	Safety is controlled at management level with use of procedure and discipline. Some training is available.	〈0.25;0.50〉
Reactive	Driven by natural instincts of being safe. Minimum fulfillment of legal standards with no/little engagement of management.	〈0;0.25〉

**Table 3 ijerph-19-02668-t003:** Cronbach alpha values for analyzed safety culture beams.

Safety Culture Beams	Cronbach Alpha	Safety Culture Beams	Cronbach Alpha
Leadership for safety	0.75	Fatal accident rate	0.74
Recognition and awards	0.71	Loss of working time	0.79
Communication and information flow	0.77	Personal injuries	0.82
Networking	0.81	Safety training	0.81
Personal knowledge and skills in field of process technology and safety	0.72	Care for others	0.74
Care for yourself	0.82	Cost of proactive actions	0.80
Loss of organization image	0.73	Organizational pride	0.79

**Table 4 ijerph-19-02668-t004:** Summary of the results of the process safety culture index (*TSC*) at organization, group, and position level, direct communication (*DI*), average communication time (*ACT*), and applicability rate of the proposed changes (*ArC*) for L, K, and G sites.

Group Position	*TSC*	*DI*	*ACT*	*ArC*
Group L	*TSC_@L_* = 0.56 (Independent)	0.44	0.32	0.58
Manager	0.53 (Independent)	0.43	0.37	0.62
Administration employee	0.64 (Independent)	0.59	0.24	0.61
Operational staff	0.52 (Independent)	0.39	0.30	0.57
Independent professional	0.64 (Independent)	0.36	0.52	0.52
Top executive	0.44 (Dependent)	1.00	0.30	0.90
Group G	*TSC_@G_* = 0.49 (Dependent)	0.33	0.26	0.73
Manager	0.58 (Independent)	0.42	0.28	0.78
Administration employee	0.52 (Independent)	0.38	0.43	0.54
Operational staff	0.46 (Dependent)	0.28	0.19	0.81
Independent professional	0.65 (Independent)	0.39	0.25	0.71
Top executive	0.63 (Independent)	1.00	0.16	0.41
Group K	*TSC_@K_* = 0.54 (Independent)	0.41	0.24	0.59
Manager	0.56 (Independent)	0.48	0.29	0.64
Administration employee	0.55 (Independent)	0.45	0.29	0.64
Operational staff	0.52 (Independent)	0.39	0.18	0.55
Independent professional	0.62 (Independent)	0.25	0.37	0.64
Top executive	0.49 (Dependent)	1.00	0.16	0.56
Average for Organization	*TSC_@O_* = 0.53 (Independent)	0.40	0.27	0.64

## Data Availability

The data presented in this study are available on request from the corresponding author upon reasonable request.

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
