# Peer review of "Methodology for the Determination of a Process Safety Culture Index and Safety Culture Maturity Level in Industries"

_ijerph, 2022, doi:10.3390/ijerph19052668_

Round 1

Reviewer 1 Report

Congratulation to the autor! Very interesting article. The culture of security in organization is a very interesting topic.

Identification of area, which need to be improved safety culture in this. Security, but mainly safety procedures implemented, and demand of them.

I consider originality to be the creation of a safety culture assessment system, a questionnaire survey, evaluation, confirmation of the hypothesis and a graphical representation of the research results.

The added value is the creation of a methodology for the implementation of security culture and a procedure for evaluating its level in the organization.  Confirmed by case study.

It is questionable whether to talk only about safety or also about security (line 91, 111,113). I think authors write about safety and security is mentioned in the article.

The results are confirmed by a significance test.

Conclusions are consistent with the evidence and arguments presented and they address the main question posed. The level of maturity of the safety culture and the way to increase it is sufficiently described in the result and discussion. References are appropriate, somwhere missing ISBN, or link to the web site, but MDPI refrence standard is very specific.

I don't have any additional comments on tables and figures

Reviewer 2 Report

The paper describes the necessity of determining safety culture and the way this has been done in Poland as a process safety culture index and an safety maturity one. The latter is the main product.

The paper is written clearly and well documented, and it is with an attractive safety maturity graphical representation. The indices are determined based on a survey.

Main comment is about the determination of safety culture. According to Prof. Dov Zohar, pioneer in the field of safety culture and safety climate, it is rather illusory to determine safety culture directly, it is only via determining safety climate that one can get an opinion about the culture. This opinion is shared by experts with deep understanding but ignored by many authors. Zohar's publications in the Journal of Applied Psychology are going back to 1980 (Dov Zohar, Safety Climate in Industrial Organizations: Theoretical and Applied Implications. Journal of Applied Psychology 1980, Vol. 65, No. 1, 96-102) 
By the way, Edgar Schein defined organizational culture (Schein, E. H. (1992). Organizational culture and leadership (2nd ed). San Francisco, Jossey-Bass). Zohar fully agrees with Schein that while leadership's attitude is crucial one cannot fathom the deepest layer in leadership's mind. This will only come out when the organization comes under pressure of time and finance. It is therefore that to make your paper also for experts more convincing that in the Introduction you describe this problem. An overview of safety culture literature is also given in the dissertation of Frank W. Guldenmund 2010, Understanding and Exploring Safety Culture. https://repository.tudelft.nl/islandora/object/uuid%3A30fb9f1c-7daf-41dd-8a5c-b6e3acfe0023?collection=research.

Visible (artifact) signs that the organization is safety aware, such as "Safety First" etc. do not say anything. Espoused values do not guarantee anything when pressure on the organization jumps up. Interviewing employees with a general questionnaire does not always help either. 

Leadership is crucial. Does leadership see the problem and wishes improvement?

Striking difference in questions to employees is that Zohar asks an employee, what a supervisor will say to him/her or will do for him/her, and not what in general the situation is as the questions you posed. According to Zohar the content of the supervisor-employee contact is crucial for safety climate (Johnson, S.E., The predictive validity of safety climate. Journal of Safety Research 38 (2007) 511–521.) Obviously, there is a relation between safety climate and culture but how is not that clear yet (Zohar, D. M., & Hofmann, D. A. (2012). Organizational culture and climate. In S. W. J. Kozlowski (Ed.), Oxford library of psychology. The Oxford handbook of organizational psychology, Vol. 1 (p. 643–666). Oxford University Press. ISBN-13: 978-0199395453).

Reference Waterson [36] in a way confirms the above by stating a theory for culture maturity models is lacking and that a weakness is the difference in results at different points of time, hence reproducibility is a point. Of course, there is uncertainty in the results which is common in any measurement. Culture does not improve without sustained efforts from the top. Hence, to see progress according to one of the maturity models needs a clear distinction with respect to criteria from a previous analysis. Therefore, I recommend the authors to make some cautionary statements in their conclusions.

Minor comment: Step 8 is not explicitly mentioned, but I assume it is describing Table 2 and Figure 2.

Reviewer 3 Report

The article sent for review concerns an important issue, which is the safety culture, and above all its shaping in enterprises.

My insights to be developed:

- In line 38-38, Authors wrote "Investigation of causes and consequences of historical and current events revealed various factors influencing positive and negative safety culture." However, the study lists the events recorded in the years 1948-2011, so they can be treated as a historical. Did Authors of the study assume a different criterion allowing the use of such a division (historical and current)?If so, it should be indicated in the study;

- Writing about positive and negative safety culture (line 39) one could define these terms. The explanation for a positive safety culture does not come until line 110;

- Line 74 - It should be indicated by whom the audit was carried out;

- Lines 105-126 - Authors refer to the safety culture and the safety climate, without indicating the relationship between them. Do these concepts are differ? Maybe they complement each other? It would be worth supplementing the information in this regard;

- What Authors understand by a “well-identified safety culture index” - line 158;

- Authors could consider introducing the chapter "Materials and Methods";

- Lines 242-243, Authors emphasize the uniqueness of the methodology. It should be stated why it is unique in comparison to other methods already described in the literature?

- Line 393 - On what basis did Authors identify 14 beams?

- The designation in Table 3 and in Figure 4,Figure 5 - Safety leadership / Leadership for safety - can be standardized;

- Results and discussion – Authors could indicate directions for future research;

- Lines 517-518 - Authors use the phrase that a relatively low-cost and easy-to-use semi-quantitative methodology for determining the total index of safety culture has been developed. Therefore, questions arise: for whom is it cheap? For whom easy to use? Since the conclusions refer to the unique experience or knowledge of the researcher;

- Lines 521-522 - what methods presented in the literature do Authors refer to?

- The references should be revised according to the IJERPH guidelines.

I hope that my insights will be useful for Authors of the study.
